# Manufacture and Vibration-Damping Effect of Composites for Archery Carbon Fiber-Reinforced Polymer Limb with Glass Fiber-Reinforced Polymer Stabilizer

**DOI:** 10.3390/ma16114048

**Published:** 2023-05-29

**Authors:** Won Wook Heo, Seung Kook An, Jeong Hyun Yeum, Seong Baek Yang, Sejin Choi

**Affiliations:** 1Department of Organic Material Science and Engineering, Pusan National University, Busan 46241, Republic of Korea; das77@daum.net; 2Department of Biofibers and Biomaterials Science, Kyungpook National University, Daegu 41566, Republic of Korea; jhyeum@knu.ac.kr; 3Research Institute for Green Energy Convergence Technology, Gyeongsang National University, Jinju 660701, Republic of Korea; sbyang@gnu.ac.kr; 4Institute of Advanced Organic Materials, Pusan National University, Busan 46241, Republic of Korea

**Keywords:** glass fiber-reinforced polymer, carbon fiber-reinforced polymer, co-curing, limb, stabilizer, archery bow, vibration-damping effect

## Abstract

Typically, archers prepare two sets of bows for competitions in case of bow breakage, but if the limbs of the bow break during a match, archers can become psychologically disadvantaged, leading to potentially fatal consequences. Archers are very sensitive to the durability and vibration of their bows. While the vibration-damping properties of Bakelite^®^ stabilizer are excellent, its low density and somewhat lower strength and durability are disadvantages. As a solution, we used carbon fiber-reinforced plastic (CFRP) and glass fiber-reinforced plastic (GFRP) for the archery limb with stabilizer, commonly used for the limbs of the bow, to manufacture the limb. The stabilizer was reverse-engineered from the Bakelite^®^ product and manufactured using glass fiber-reinforced plastic in the same shape as the existing product. Analyzing the vibration-damping effect and researching ways to reduce the vibration that occurs during shooting through 3D modeling and simulation, it was possible to evaluate the characteristics and the effect of reducing the limb’s vibration by manufacturing archery bows and limbs using carbon fiber- and glass fiber-reinforced composites. The objective of this study was to manufacture archery bows using CFRP and GFRP, and to assess their characteristics as well as their effectiveness at reducing limb vibration. Through testing, the limb and stabilizer that were produced were determined to not fall behind the abilities of the bows currently used by athletes, and they also exhibited a noticeable reduction in vibrations.

## 1. Introduction

Fiber-reinforced composites have been widely used in sports, energy, transportation and civil engineering, owing to their superior mechanical, fatigue, and durability performances [1,2,3,4]. Composites play a crucial role in the manufacturing of sports equipment. For instance, in archery and arrow sports, composite materials are commonly used to make the archery limb and stabilizer. These materials have high strength and lightweight properties, making them extremely useful for enhancing the performance of sports equipment [5,6,7,8]. These materials possess vibration-damping properties, which play a significant role in improving the stability of sports equipment [9]. This feature is especially useful for sports equipment used in high-speed and high-vibration sports. For these reasons, prepreg composites are integral to the manufacturing of sports equipment, providing high performance, durability, and stability, and helping athletes achieve better results [10,11].

Vibrations are crucially important in archery. Archery is a precision sport that requires very accurate techniques, and vibrations from the bow and arrows can have a significant impact on aiming and accuracy. Vibrations can reduce the stability of the bow and arrows and create instability in the aiming and hitting position. Therefore, minimizing the vibrations in archery equipment is essential to improve performance and accuracy. Archery equipment manufacturers are developing various technologies and solutions to reduce vibrations and improve performance [12].

Figure 1a illustrates the terminology of each part of the bow, where the limb (Figure 1b) and stabilizer (Figure 1c) of the bow serve different roles. The limb, located in the center of the bow, is responsible for tensioning the bow. When the bow is tensioned, the limb bends and becomes elastic, and when the tension is released, the limb returns to its original shape. This movement stores energy in the bow, which is then used to shoot an arrow. The stabilizer, a component attached to the end of the bow, is responsible for reducing vibrations [13]. When the bow is fired, the limb releases a large amount of energy and produces vibrations, which can affect aiming and accuracy. The stabilizer reduces this vibration, increasing the stability of the bow, and improving aiming and accuracy. Additionally, the stabilizer can be used to adjust the balance of the bow by changing its center of gravity.

Carbon fiber-reinforced plastic (CFRP) and glass fiber-reinforced plastic (GFRP) are composites made by combining fibers, such as carbon or glass, with a resin matrix. CFRP has even higher strength, stiffness, and low density, as well as high resistance to acids and corrosion, making it popular in industries such as aerospace, automotive, and sports equipment [14,15]. In contrast, GFRP is relatively inexpensive and lightweight, with high durability and corrosion resistance, making it commonly used in construction, automotive parts, boats, and sports equipment [16,17]. However, its high production cost is a drawback. Recently, CFRP has been extensively researched in various fields due to its lightweight and energy-saving properties [18].

Co-curing is a composite manufacturing technique where multiple layers of materials, such as fibers and resin, are cured simultaneously under the same processing conditions [19,20]. Austermann et al. studied the fiber-reinforced composite sandwich structures by co-curing with additive manufactured epoxy lattices [21]. This method involves placing all the materials, including reinforcements and resins, in a mold and curing them together. This technique offers advantages such as improved inter-laminar shear strength, reduced voids, and better overall mechanical properties of the final composite product. The process is often used in the aerospace industry for manufacturing large, complex structures, as it helps to reduce manufacturing time and costs while producing high-quality composites.

The purpose of this study is to compare and analyze composites for archery limbs using CFRP from three different companies to identify any differences and evaluate their performance. During the study, issues with delamination causing fatigue strength were found in the manufacturing of CFRP limbs and in attaching Bakelite^®^ stabilizers to existing limbs. Bakelite^®^ stabilizers were of interest in this study due to their vibration-damping properties and use of autoclave co-cure molding techniques. This study aimed to determine whether replacing Bakelite^®^ stabilizers with GFRP stabilizers would reduce vibrations, and further experiments were conducted to test this hypothesis. Overall, this study analyzed the properties of various composites for archery limbs and identified the potential issues related to the application and vibration-damping effects of newly manufactured composites for archery limbs.

## 2. Materials and Methods

### 2.1. Materials

Unidirectional (UD) carbon fiber (Toray, Japan) was used to manufacture limbs using prepreg products from various companies (Hankook Carbon Co., Ltd. (HK), Milyang, Republic of Korea; TB Carbon Co., Ltd. (TB), Gimhae, Republic of Korea; SK Chemical Co., Ltd. (SK), Seong-Nam, Republic of Korea). The prepregs had a resin content of 38% and were loaded with a 200 g weight. For the limb, a prepreg product woven using Mitsubishi Rayon Co., Ltd.’s carbon fiber in a plain weave pattern from Hyundai Fiber (HF) Co., Ltd, (Yang-san, Republic of Korea). was used. The carbon fiber prepreg had a resin content of 42%. Glass fiber (PPG industries, Taiwan) was used to produce the stabilizer for this study, and Hyundai Fiber (HF2) Co., Ltd.’s 7628 fabric prepreg (Yang-san, Republic of Korea), coated using the solvent technique, was used to fabricate the GFRP (Table 1). The existing recurve bow limbs for athletes all use carbon 3k fabric on the skin surface and carbon U/D on the inside. They are manufactured to the same specifications as those used in actual international competitions. The reason for using GFRP is that it has excellent adhesion and moldability when co-cured with carbon, and the finished product is easy to process with the added benefit of the ability to easily adjust the thickness, color, and material according to the needs.

### 2.2. Manufacturing of CFRP Composite for Limbs

All materials, including molding auxiliary materials, were placed in a vacuum bag inside an autoclave to create a composite specimen. Two sheets of UD carbon prepregs (45°, 500 × 500 mm^2^) prepared in plain weave were laminated to the outermost layer after the SUS mold had undergone a release treatment before being placed in the autoclave. Between the 2 sheets, 90° of HK, TB, and SK’s UD carbon prepreg were applied. In the orientation shown in Figure 2, 12 sheets were stacked. The laminated prepreg was placed in the proper order on the outside, molded for 120 min at 950 MPa vacuum pressure and approximately 8 kg/cm^2^ pneumatic pressure, and the temperature inside the autoclave was gradually lowered. A carbon fiber composite sample was prepared after the temperature dropped to around 30 °C using cooled water for 40 min.

It is known that the volume ratio of carbon fibers (FVF) in the composites has a great influence on the mechanical properties of the composites. The theoretical FVF in the prepared specimen was calculated using the following formula:$$v_{f}=\frac{FAW\times n}{\rho_{f}\times t}\times 100$$
where *v_f_* (%): volume-to-volume fiber content; *FAW* (g/cm^2^): fiber areal weight; *n*: number of plied sheets; *ρ_f_* (g/cm^3^): density of carbon fiber; and *t* (mm): thickness of samples. Specimens were manufactured using prepreg under the same conditions, and the experiment was conducted.

### 2.3. Manufacturing of Limb with Stabilizer

Based on the ASTM standard, a test specimen was manufactured and tested, and the #C2 product showed the enhanced results. Using this result, #C2 prepreg was laminated at 90° direction between HF carbon 3k plain prepreg, and two maple leaves were inserted between the UD prepreg layers and bonded with epoxy bond. It was fabricated by a limb for a bow that was specifically designed for vibration analysis. Bakelite^®^ material (supplied by Haodesheng Insulation Material in China) and HF2 (7628 fabric) were laminated on the outer surface of the 3k plain prepreg at the stabilizer area, and a co-curing method was applied to mold all materials at once using autoclave molding, resulting in the production of an archery limb and stabilizer. CFRP and GFRP were stacked in prepreg form in a mold and molded together in one step using co-curing in an autoclave.

### 2.4. Design of Limb with Stabilizer

To compare with existing commercial bows, the dimensions of a 2D-scanned and modeled bow limb were used to design the archery bow limb shown in Figure 3a. Using a 3D scanner, the dimensions of a bow limb were reverse-engineered, and based on that shape, the upper and lower molds were designed in computer-aided design (CAD), shown in Figure 3b,c. The bow limb consists of three main parts: the stabilizer part that attaches to the handle, the flat part, and the curved part.

### 2.5. Characterizations

Scanning electron microscopy (SEM, SUPRA25, Carl Zeiss AG, Jena, Germany) was performed to confirm the morphological characteristics. Each prepreg prepared with carbon fiber from different companies (HK, TB, and SK) had its cross-section measured. Using a universal testing machine (UTM 5882, Instron, MA, USA), the manufactured composites’ mechanical characteristics were evaluated. ASTM D638 standards were used to measure the tensile properties; ASTM D790 standards were used to measure the flexure properties; and ASTM D695 standards were used to measure the compressive values. To manufacture the test specimens, 5 were produced for each group, and an experiment was carried out. To compare the durability of Bakelite^®^ stabilizer and GFRP stabilizer, tensile strength was measured using ASTM D638 standard on dog-bone-type samples fabricated by Bakelite one and GFRP, and tested at a speed of 5 mm/min using UTM 5882. Flexural strength was also measured using Instron’s UTM 5882 according to ASTM D790 standard, using 3-point loading method at a speed of 2 mm/min. The compressive strength of the samples was measured in accordance with ASTM D695 using UTM 5882 at a compression speed of 1.3 mm/min.

### 2.6. Simulation of Limb Shooting Process

Using the Catia program, structural analysis simulations were conducted to design and analyze the shape of the bow limb. The finite element method (FEM) modeling was created by scanning and modeling the shape of the Win&Win archery bow limb, which is the most widely used model among athletes. In the experiment, the bow limb and stabilizer were attached to the handle, similarly to the existing archery bow. The bow was then shot under different conditions, applying a force in the Y direction toward the curved end of the bow limb, pulling it approximately 50–100 mm. The 3D shape was then analyzed after shooting.

### 2.7. Vibration-Damping Test of Manufactured CFRP Limb with Stabilizer

The vibration test (Figure 4) measured the results at 25 °C with vibration receiver equipment (TSVA-PRO, Signalling Co., Ltd., Seoul, Republic of Korea) and EASTON Co., Ltd.’s arrows (California City, CA, USA). In the X, Y, and Z axes, the vibrations produced by shooting an arrow were measured. The limb used for test analysis was employed in this study, and the arrow handle used in the experiment was one made by Win&Win Co., Ltd., (Ansung, Republic of Korea). Additionally, the vibration test was carried out 4 times with the freshly created 68.42 lb. archery limb under the same circumstances as the prior limb experiment.

## 3. Results and Discussion

As a result of calculating the FVF manufactured using various carbon fibers according to the equation, #C2 was measured the highest (Figure 5a). The properties of composites depend on various factors, such as the type, amount, and orientation of reinforcement fibers; the type and amount of matrix material; the manufacturing process; and the quality of the interface between the fibers and the matrix. In the case of mechanical properties, for example, the stiffness and strength of a composite depend primarily on the type, orientation, and volume fraction of the fibers, as well as the quality of the fiber–matrix interface. Generally, stiffer and stronger fibers result in stiffer and stronger composites, while a better fiber–matrix interface can enhance load transfer and prevent premature failure. Additionally, as a result of measuring the specimen weight and resin contents of the prepared prepreg, it was confirmed that the resin content of #C2 was measured to be the lowest. The low resin content means that the number of voids generated inside the composites is the smallest during the prepreg process, which is thought to affect the mechanical properties [22,23,24]. In addition, since the weight of the specimen is light, if it exhibits the same mechanical performance, it is expected that the lighter one will be suitable for use as a composite.

During the production process of carbon prepregs, the volatile solvent used in the composites’ epoxy resin cannot escape to the outside during the curing process and remains inside, leaving many residual bubbles. The amount of residual bubbles left behind can have a significant impact on the properties of the composites. Unlike general prepreg fabrics, UD prepreg is stacked in the shape of each UD, not the weave pattern of the fabric, making it practically impossible to remove all voids that occur between layers during autoclave molding. Additionally, depending on the ingredients used by each company producing epoxy resin, it is also possible for traces of voids to remain inside the test specimen due to the inability to completely remove volatile substances [25,26]. It is confirmed that all the specimens in Figure 6a–c formed composites by impregnating epoxy resin into the carbon fiber prepreg. However, in Figure 6, SEM images reveal that the processed surface of product #C1 contains numerous pinholes, while products #C2 and #C3 exhibit a clean cross-section.

Figure 7 presents the results of the mechanical property measurements of carbon fiber prepregs manufactured by various companies. Figure 7a shows the results for tensile properties, and all three specimens exhibited a similar initial trend. In terms of flexural strength, #C2 showed superior results (Figure 7b), while in terms of compressive strength, #C2 exhibited better results than the other specimens. For tensile modulus, #C3 showed the best performance, but in terms of flexural modulus and compressive modulus, #C2 exhibited the highest values. Therefore, it is considered appropriate to use the SK, which is the manufacturer of #C2, for fabricating archery limbs.

The mechanical properties of the GFRP and the Bakelite^®^ stabilizers currently on the market were measured, and the results are shown in Figure 8. Figure 8a shows the results of the tensile properties, and GFRP exhibited superior strength, modulus, and elasticity compared to Bakelite^®^. In terms of bending strength, GFRP showed higher strength and modulus than Bakelite^®^ (Figure 8b). This is because Bakelite’s stabilizer can bend easily even with a small force when the bow is pulled, which may lead to reduced stability for the user. For compressive strength, Bakelite^®^ showed better compressive strain and GFRP exhibited higher strength.

A composite’s bow limb for archery will be developed using a carbon UD prepreg and 3K prepreg, along with Bakelite^®^ stabilizers. The limb will be produced using autoclave molding and will feature new GFRP stabilizers in place of the previous Bakelite-based design. These stabilizers are expected to enhance the limb’s performance. The materials will be based on scanned shape data, using carbon with a Young’s modulus of 1.35 × 10^11^ N·m^−2^, a Poisson’s ratio of 0.46, a density of 1600 kg/m^3^, and a yield strength of 1.3 × 10^9^ N·m^−2^. In the experiment, the testing process will involve applying the same conditions as when shooting an actual arrow, which includes restricting the limb’s stabilizer to the handle and pulling it approximately 100 mm in the Y direction toward the curved tip. The modeling of the load generated when the bowstring is pulled halfway (about 50 mm) will be demonstrated, as well as the modeling of the load generated when the bowstring is fully pulled (about 100 mm), and the load characteristics generated on the archery bow limb after shooting an arrow. Based on the 3D modeling, it can be confirmed that the load applied to the bow limb is all in the same position until the moment of shooting the arrow by pulling the bowstring. Looking at the vibration-damping modeling, during the moment of shooting the arrow by pulling the bowstring all the way (approximately 100 mm), the bending load is transferred to the tip of the limb, located at the curved tip of the stabilizer, due to the damping of the vibration generated during shooting. It can be confirmed through 3D simulation as it can be shown in Figure 9.

Upon examining the simulation results in Table 2 and Table 3, it can be seen that the vibration at the curved tip of the limb, which occurs when shooting the bow after pulling the string to the full draw length (about 100 mm), is smaller in the case of the GFRP stabilizer applied to the archery bow compared to the Bakelite^®^ stabilizer. However, it was observed that the vibration transmitted to the stabilizer and causing oscillation was greater in the case of the GFRP material compared to the Bakelite^®^. After analyzing the vibration data, it can be seen that the amplitude of the inherent vibration is greater for the GFRP stabilizer, but when the vibration is damped, it causes a smoother curve compared to the Bakelite^®^ stabilizer. Additionally, it can be observed from the vibration graph that the GFRP stabilizer draws the waveforms with smoother peaks and troughs compared to the Bakelite^®^ stabilizer.

The vibration generated during shooting of limbs equipped with Bakelite^®^ stabilizers was represented, and the final vibration suppression was shown in Figure 10. In this figure, the T direction represents the vibration generated at the curved tip during shooting, with the amplitude in the T_x_ direction being the highest at 3.20% at 38.4368 Hz, while the amplitude in the T_y_ direction being the highest at 25.30% at 126.87 Hz. The amplitude in the T_z_ direction is the highest at 28.36% at 38.4368 Hz. The R direction absorbs the vibration generated in the T direction at the Bakelite^®^ stabilizer and cancels it out. The amplitude in the R_x_ direction is the highest at 61.99% at 126.87 Hz, while the amplitude in the R_y_ direction is the highest at 29.96% at 38.437 Hz. The amplitude in the R_z_ direction is the highest at 24.26% at 126.87 Hz. It was observed that the magnitude of the vibration from the curved tip toward the Bakelite^®^ stabilizer differed in each direction when shooting with archery limbs. As the vibration in the T_z_ direction decreased from 28.36% to 24.26% while moving toward the R_z_ direction, the vibration in the T_y_ direction increased from 25.30% to 29.96%, and the vibration in the T_x_ direction increased from 3.20% to 61.99% as it moved toward the R_x_ direction. The 3D simulation results showed that the vibration in the T_x_ direction, which is the vertical vibration generated during shooting, was amplified significantly from 6.58% to 73.57% as it was transmitted toward the R_x_ direction. The vibration in the T_y_ direction, which is the front-to-back vibration, decreased by about 50% from 69.86% to 32.31% as it was transmitted toward the R_y_ direction, and the vibration in the T_z_ direction, which is the left-to-right vibration, decreased by about 52% from 65.80% to 29.50% as it was transmitted toward the R_z_ direction.

According to the measurement results of the vibration that occur during shooting of limbs equipped with GFRP stabilizers, the T direction exhibits the largest amplitude in the T_x_ direction at 39.3782 Hz, accounting for 2.96% of the total, and in the T_y_ direction at 42.665 Hz, accounting for 19.89% of the total. Meanwhile, in the T_z_ direction, the largest amplitude occurs at 39.3782 Hz, accounting for 24.46% of the total. The R direction absorbs the vibration generated in the T direction at the GFRP stabilizer and cancels it out. The largest amplitude in the R_x_ direction occurs at 13.553 Hz, accounting for 69.62% of the total, while the largest amplitude in the R_y_ direction occurs at 39.3782 Hz, accounting for 34.76% of the total. The largest amplitude in the R_z_ direction occurs at 13.553 Hz, accounting for 26.53% of the total. As shown in Table 3, it was observed that the size of the vibration at each direction is transferred differently from the curved tip to the GFRP stabilizer when shooting arrows with archery limbs. The T_z_ direction decreased from 57.60% to 33.22% as it moved toward the R_z_ direction, while the T_y_ direction decreased from 47.37% to 36.45% as it moved toward the R_y_ direction. The T_x_ direction exhibited a larger vibration, increasing from 6.10% to 33.22% toward the R_x_ direction. The 3D simulation results revealed that the vibration in the T_x_ direction, which occurs in the upward and downward directions during shooting, is amplified from 6.10% to 33.22% when transmitted to the R_x_ direction. The vibration in the T_y_ direction, which occurs in the front and back directions, decreases by approximately 25% as it is transmitted to the R_y_ direction, reducing from 47.37% to 36.45%. Finally, the vibration in the T_z_ direction, which occurs in the left and right directions, decreases by approximately 20% as it is transmitted to the R_z_ direction, reducing from 57.60% to 47.37%.

To reduce the vibration in the X and Y axes, it is intended to use a material with a higher density, GFRP stabilizer, instead of Bakelite^®^ stabilizer, which is currently used for the stabilizer that connects the existing bow limbs and handle. The vibration test results showed a damping effect on vibration in the X axis, resulting in stable vibration. However, there was a tendency for less improvement in the Y and Z-axes compared to the *X* axis. When using the GFRP stabilizer, it was observed that there was an improvement of approximately 50% in the X axis, approximately 10% in the Y axis, and approximately 30% in the *Z* axis compared to when using the Bakelite^®^ stabilizer (Figure 11).

Using the newly manufactured 68.42 lb. bow limbs with a GFRP stabilizer applied, we conducted vibration tests under identical conditions four times, as in the previous experiment, and the results for the X, Y, and Z axes are shown in Figure 12. The thickness of the GFRP stabilizer that was previously applied was the same as that of the Bakelite^®^, at 1.5 mm. However, we reduced the thickness of the GFRP stabilizer to 1 mm to reduce the weight of the limbs and confirm the absorption of vibrations. When the vibration test results were examined with the 1 mm GFRP stabilizer thickness, it was seen that the vibration in the X, Y, and Z axes increased by 10, 100, and 150 m·s^−2^, respectively, compared to the previous Bakelite^®^ stabilizer. However, the vibration did not increase significantly compared to the limbs with the previous 1.5 mm GFRP stabilizer.

## 4. Conclusions

In the world of archery, the limbs and stabilizers of athletes’ bows are commonly made of Bakelite^®^ material. However, the downside of Bakelite^®^ is that it loses durability over time and can experience delamination. To address this issue, researchers have applied a high-density, easy-to-process, and bonding-effective GFRP material, commonly used in carbon prepreg and bonding for traditional bow limbs, to the limbs and stabilizers. The goal was to determine if the newly developed stabilizer could replace Bakelite’s one. Using various companies’ carbon prepreg materials, the #C2 product showed the best results in all experiments except for tensile strength. Before producing the actual archery bow limbs, researchers scanned and created a 3D modeling file of Win&Win’s bow limbs using a 3D scanner and used this data to verify if the stabilizer could be replaced from Bakelite^®^ to GFRP using 3D simulation. In actual shooting vibration tests, limbs with Bakelite^®^ stabilizer, 1.5 mm GFRP stabilizer, and 1 mm GFRP stabilizer were produced and analyzed. The results showed that the limbs with Bakelite^®^ stabilizer exhibited an average vibration of 500, 2000, and 1200 m·s^−2^ on the X, Y, and Z axes, respectively. In contrast, the limbs with 1.5 mm GFRP stabilizer showed a significantly reduced vibration of 280, 1500, and 800 m·s^−2^ compared to Bakelite^®^ stabilizer. Additionally, limbs with 1 mm GFRP stabilizer showed slightly higher vibration values of 290, 1600, and 950 m·s^−2^ than those with 1.5 mm GFRP stabilizer, but they showed a reduction in vibration of approximately 42% on the X axis, 20% on the Y axis, and 21% on the Z axis compared to those with Bakelite stabilizer. This study confirms that GFRP stabilizers are much more vibration-resistant and cause less vibration than Bakelite^®^ stabilizer in archery bow limbs. On average, vibration decreased by approximately 45%, 25%, and 33% in the X, Y, and Z axes, respectively, based on objective numerical data. However, the quality of a product cannot be evaluated solely based on the quality of the carbon and the manufacturing process, as the success of a sport also depends on the athlete’s ability to use the equipment effectively.

## Figures and Tables

**Figure 1 materials-16-04048-f001:**
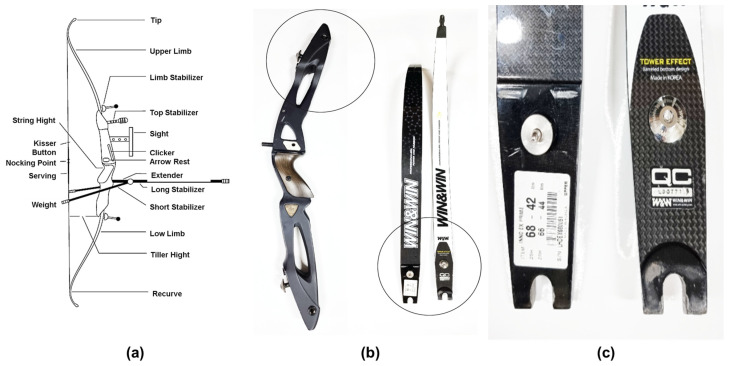
(**a**) Bow parts terminology; (**b**) photograph depicting the limbs of the bow; and (**c**) photograph showing the stabilizer of the bow.

**Figure 2 materials-16-04048-f002:**
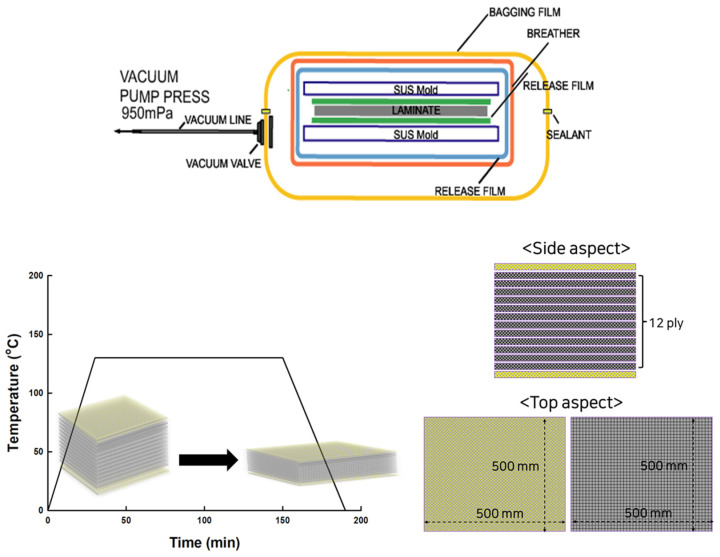
Schematic illustration of autoclave processing, process conditions, and lamination conditions.

**Figure 3 materials-16-04048-f003:**
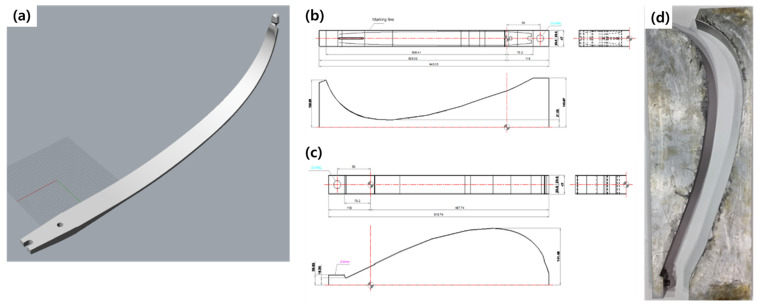
(**a**) Three-dimensional graphic image of manufactured limb. CAD drawing of (**b**) lower and (**c**) upper molds for archery limb based on 3D scanner data. (**d**) Photograph of aluminum limb mold.

**Figure 4 materials-16-04048-f004:**
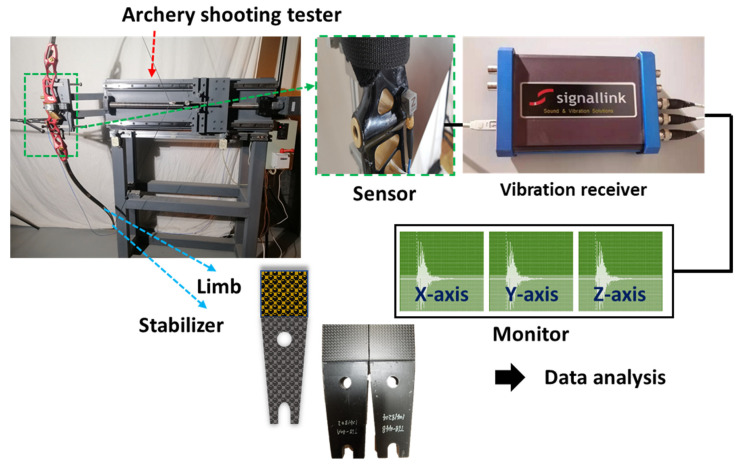
Photograph of test devices and process of vibration-damping test.

**Figure 5 materials-16-04048-f005:**
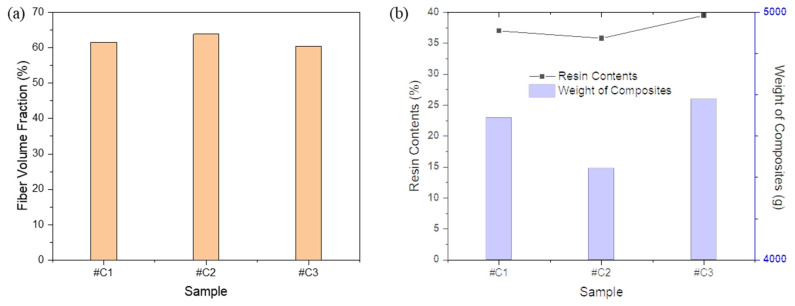
(**a**) FVF depending on sample type; (**b**) resin content and weight of composites according to sample type.

**Figure 6 materials-16-04048-f006:**
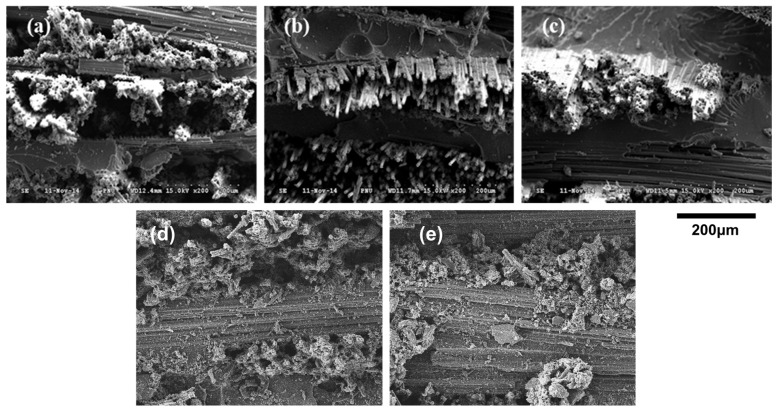
SEM images of composites fabricated with (**a**) #C1, (**b**) #C2, (**c**) #C3, (**d**) #G1, and (**e**) #G2.

**Figure 7 materials-16-04048-f007:**
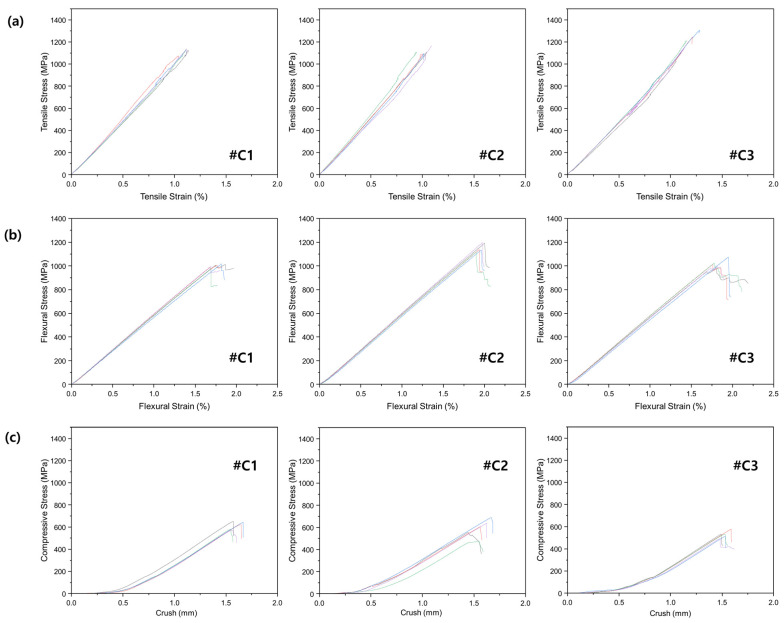
Mechanical properties of carbon fiber prepreg manufactured by various companies (#C1: HK, #C2: SK, and #C3: TB). (**a**) Tensile property; (**b**) flexural property; and (**c**) compression property.

**Figure 8 materials-16-04048-f008:**
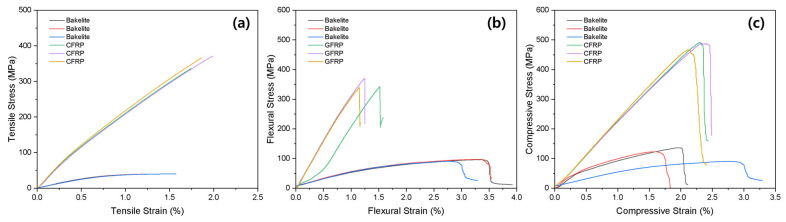
(**a**) Tensile, (**b**) flexural, and (**c**) compressive properties of Bakelite^®^ and GFRP stabilizers.

**Figure 9 materials-16-04048-f009:**
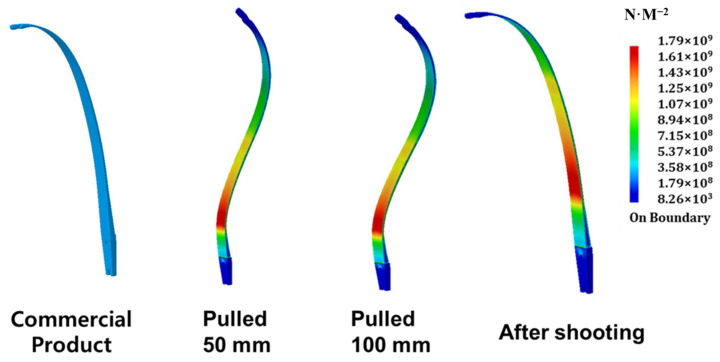
Scanning simulation comparison of commercial product, pulled 50 mm and 100 mm, and after shooting limbs with stabilizer.

**Figure 10 materials-16-04048-f010:**
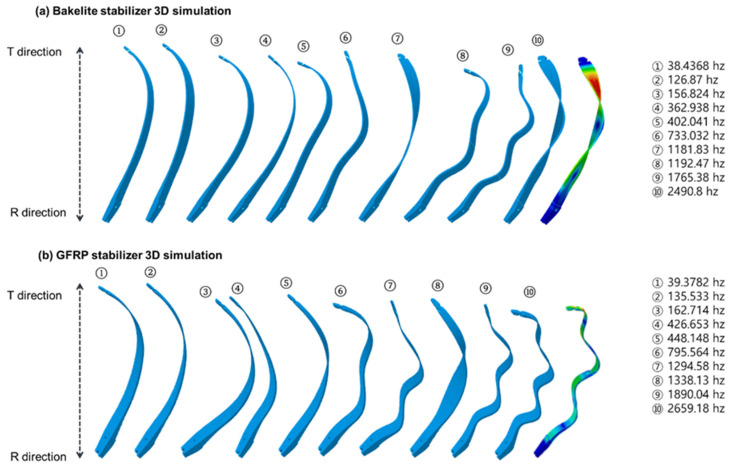
3D simulation of limb movement with (**a**) Bakelite stabilizer and (**b**) GFRP stabilizer depending on various frequency.

**Figure 11 materials-16-04048-f011:**
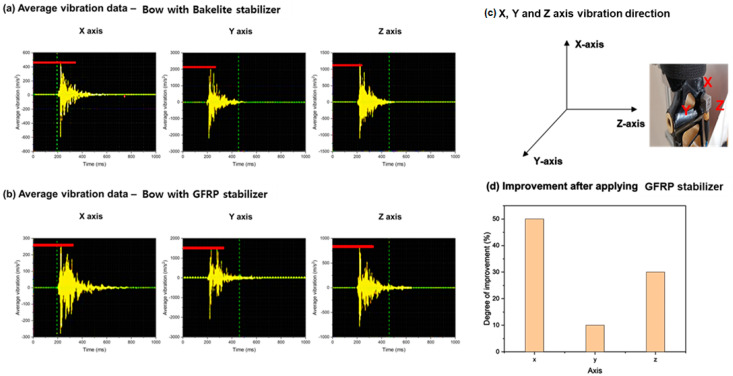
(**a**) Average vibration of bow with Bakelite stabilizer depending on various axes, (**b**) average vibration of bow with GFRP (#G1) stabilizer, (**c**) vibration direction and (**d**) improvement after applying GFRP (#G2) stabilizer.

**Figure 12 materials-16-04048-f012:**
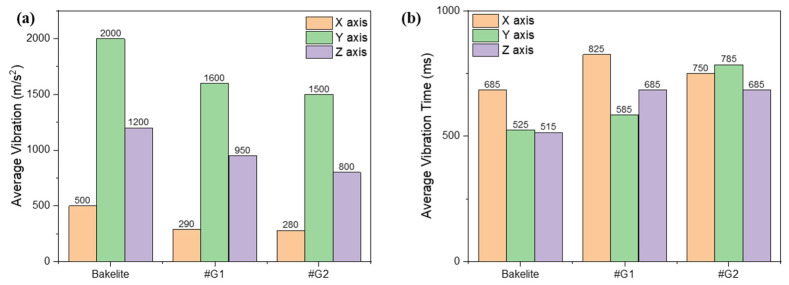
(**a**) Average vibration and (**b**) average vibration time of bow with Bakelite stabilizer and GFRP stabilizer (#G1, #G2) depending on various axes.

**Table 1 materials-16-04048-t001:** The thickness and shorthand notation of the carbon fiber fabric used for the limb and the glass fiber fabric used for the stabilizer.

Parts	Materials	Company	Thickness (T)	Sample Name
Limb	Carbon Fiber	HK	2.8	#C1
Limb	Carbon Fiber	TB	3.0	#C2
Limb	Carbon Fiber	SK	3.2	#C3
Stabilizer	Glass Fiber	HF	1.0	#G1
Stabilizer	Glass Fiber	HF	1.5	#G2

**Table 2 materials-16-04048-t002:** The amplitude of Bakelite^®^ stabilizer depending on frequency.

No.	Frequency (Hz)	T_x_ (%)	T_y_ (%)	T_z_ (%)	R_x_ (%)	R_y_ (%)	R_z_ (%)
1	38.437	3.2	0.0	38.36	0.0	29.96	0.0
2	126.87	0.0	25.3	0.0	61.99	0.0	24.26
3	156.82	2.93	0.0	16.2	0.0	0.11	0.0
4	362.94	0.0	23.38	0.0	5.48	0.0	2.72
5	402.04	0.0	0.0	8.28	0.0	0.28	0.0
6	733.03	0.28	0.0	6.39	0.0	0.58	0.0
7	1181.8	0.0	8.83	0.0	4.48	0.0	1.03
8	1192.5	0.0	0.0	4.1	0.0	0.72	0.0
9	1765.4	0.17	0.0	2.47	0.0	0.65	0.0
10	2490.8	0.0	5.35	0.0	1.61	0.0	1.48

**Table 3 materials-16-04048-t003:** The amplitude of GFRP stabilizer depending on frequency.

No.	Frequency (Hz)	T_x_ (%)	T_y_ (%)	T_z_ (%)	R_x_ (%)	R_y_ (%)	R_z_ (%)
1	39.378	2.96	0.0	24.46	0.0	34.76	0.0
2	135.53	0.0	18.69	0.0	68.62	0.0	26.53
3	162.71	2.52	0.0	13.42	0.0	0.57	0.0
4	426.65	0.0	0.0	6.05	0.0	0.01	0.0
5	448.15	0.0	19.89	0.0	1.26	0.0	6.64
6	795.56	0.31	0.0	4.88	0.0	0.11	0.0
7	1294.6	0.08	0.0	3.76	0.0	0.25	0.0
8	1338.1	0.0	8.79	0.0	3.1	0.0	0.06
9	1890.0	0.03	0.0	2.95	0.0	0.36	0.0
10	2659.2	0.2	0.0	2.07	0.0	0.38	0.0

## Data Availability

Not applicable.

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
