# Peer review of "Manufacture and Vibration-Damping Effect of Composites for Archery Carbon Fiber-Reinforced Polymer Limb with Glass Fiber-Reinforced Polymer Stabilizer"

_materials, 2023, doi:10.3390/ma16114048_

Round 1
Reviewer 1 Report
1. Authors should use full names for CFRP and GFRP in the title.
2. Authors should include main findings in the abstract.
3. Fonts in Figure 1a are too small.
4. Can authors provide rationales why CF and GF were used for a limb and a stabilizer, respectively?
5. Please include countries for all materials and equipment used in this work.
6. Heading for section 2.3 and 2.4 are the same.
7. I would like to see more discussion on why different composites exhibited different properties, especially mechanical properties.
8. Is there any referenced or requirement for actual equipment to compare?
Minor changes are required.
Author Response
Dear Editor, Miss Raluca Stefan
Thank you very much for your message of April 17th, 2023 regarding our manuscript. We also would like to thank you for sending the reviewer’s comments to us. Revised manuscript and the list of corrections have been set as per your suggestion. Our incorporation of the reviewer’s suggestion is as follows:
1. Authors should use full names for CFRP and GFRP in the title.
Answer: We corrected the title by using the full names of CFRP and GFRP.
2. Authors should include main findings in the abstract.
Answer: We have confirmed that the manufactured CFRP limb and GFRP stabilizer exhibit no performance inferiority to the bows currently used by athletes, and have a vibration reduction effect. We add the last line: “Through testing, the limb and stabilizer that were produced were determined to not fall behind the abilities of the bows currently used by athletes, and they also exhibit a noticeable reduction in vibration.”
3. Fonts in Figure 1a are too small.
Answer: We have replaced the corrected one.
4. Can authors provide rationales why CF and GF were used for a limb and a stabilizer, respectively?
Answer: The existing recurve bow limbs for athletes all use carbon 3k fabric on the skin surface and carbon U/D on the inside. (They are manufactured to the same specifications as those used in actual international competitions.) The reason for using GFRP is that it has excellent adhesion and moldability when co-cured with carbon, and the finished product is easy to process with the added benefit of the ability to easily adjust the thickness, color, and material according to the needs.
5. Please include countries for all materials and equipment used in this work.
Answer:
Carbon 3k twill prepreg: The carbon fiber is from Toray Industries in Japan, and the carbon weaving and prepreg work are done by Hyundai-WIA in Korea.
Carbon U/D: The carbon fiber is from Toray Industries in Japan, and the prepreg work is done by SK KAKEN in Korea.
GFRP: The glass fiber is from PPG Industries in Taiwan, and the weaving and prepreg work are done by Hyundai-WIA in Korea.
Bakelite: Supplied by HAODESHENG INSULATION MATERIAL in China
6. Heading for section 2.3 and 2.4 are the same.
Answer: We revised the heading like “Design of limb with stabilizer”
7. I would like to see more discussion on why different composites exhibited different properties, especially mechanical properties.
Answer: we added the comment. “The properties of composites depend on various factors such as the type, amount, and orientation of reinforcement fibers, the type and amount of matrix material, the manufacturing process, and the quality of the interface between the fibers and the matrix. In the case of mechanical properties, for example, the stiffness and strength of a composite depend primarily on the type, orientation, and volume fraction of the fibers, as well as the quality of the fiber-matrix interface. Generally, stiffer and stronger fibers result in stiffer and stronger composites, while a better fiber-matrix interface can enhance load transfer and prevent premature failure.”
8. Is there any referenced or requirement for actual equipment to compare?
Answer: We compared using real products. In order to see the vibration damping effect, we compared it with the Win&win bow product equipped with Bakelite's Stabilizer. (comparison with actual product)

Reviewer 2 Report
In this work by Won Wook Heo, Seung Kook An, Jeong Hyun Yeum, Seong Baek Yang and Sejin Choi entitled “Manufacture and Vibration Damping Effect of Composites for 2 Archery CFRP Limb with GFRP Stabilizer”, the authors evaluate the mechanical properties of 3 kinds of carbon fiber prepreg and 2 kinds of glass fiber prepreg were tested by tensile, bending and compression tests, and the fiber materials which can be used to prepare limbs and stabilizers were obtained; use autoclave molding to form all materials in one go to create the arrow limb and stabilizer and the effects of bakelite stabilizer and GFRP stabilizer are compared by finite element simulation technique and vibration test. This work provides a material choice for manufacture archery bows. But there are the following problems:
1. In this paper, the damping effect of bakelite stabilizer and GRFP stabilizer is simply stated through vibration test, without theoretical analysis of the damping results, and there is a lack of innovation in preparation method and finite element analysis.
2. The experimental data of tensile, bending and compression tests at material level are few and not universal.
3. The title of 2.3 and 2.4 are the same.
4. It can be seen from Fig. 8b that the bending modulus of GFRP stabilizer is significantly higher than that of bakelite stabilizer, which is inconsistent with the author's description.
5. Fig.7 and Fig.8 are not in the same format.
6. Table 2, Table 3 and Table 1 are in different formats.
7. Directions X, Y and Z should be marked on the model in the instructions in Fig.11
none
Author Response
Dear Editor, Miss Raluca Stefan
Thank you very much for your message of April 17th, 2023 regarding our manuscript. We also would like to thank you for sending the reviewer’s comments to us. Revised manuscript and the list of corrections have been set as per your suggestion. Our incorporation of the reviewer’s suggestion is as follows:
In this work by Won Wook Heo, Seung Kook An, Jeong Hyun Yeum, Seong Baek Yang and Sejin Choi entitled “Manufacture and Vibration Damping Effect of Composites for 2 Archery CFRP Limb with GFRP Stabilizer”, the authors evaluate the mechanical properties of 3 kinds of carbon fiber prepreg and 2 kinds of glass fiber prepreg were tested by tensile, bending and compression tests, and the fiber materials which can be used to prepare limbs and stabilizers were obtained; use autoclave molding to form all materials in one go to create the arrow limb and stabilizer and the effects of bakelite stabilizer and GFRP stabilizer are compared by finite element simulation technique and vibration test. This work provides a material choice for manufacture archery bows. But there are the following problems:
- In this paper, the damping effect of bakelite stabilizer and GRFP stabilizer is simply stated through vibration test, without theoretical analysis of the damping results, and there is a lack of innovation in preparation method and finite element analysis.
Answer: Thank you for your insightful comment. It seems that we focused more on the practical aspect of vibration damping and its role in sports equipment rather than the complex theoretical aspects in this paper. In future research, considering the theoretical aspects as you have pointed out would be an interesting topic to explore.
- The experimental data of tensile, bending and compression tests at material level are few and not universal.
Answer: We have supplemented Figures 7 and 8 with experimental data from tensile, bending, and compression tests.
- The title of 2.3 and 2.4 are the same.
Answer: We revised the heading like “Design of limb with stabilizer”
- It can be seen from Fig. 8b that the bending modulus of GFRP stabilizer is significantly higher than that of bakelite stabilizer, which is inconsistent with the author's description.
Answer: We have revised it to: "In terms of bending strength, GFRP showed higher strength and modulus than Bakelite® (Fig. 8b). This is because Bakelite® has a tendency to bend easily even with a small force when the bow is pulled, which may compromise the user's stability. For compressive strength, Bakelite® exhibited better compressive strain while GFRP showed higher strength."
- Fig.7 and Fig.8 are not in the same format.
Answer: I have formatted it correctly
- Table 2, Table 3 and Table 1 are in different formats.
Answer: I have formatted it correctly
- Directions X, Y and Z should be marked on the model in the instructions in Fig.11
Answer: I have attached a picture with the appropriate labeling for the X, Y, and Z axes

Reviewer 3 Report
Accepted with minor revisions
a)to Fig. 6 for SEM images (It's better to present the dimensions I think it is 200μm for comparison and put out the notes)
b)SEM images for G1 and G2?
c)in Figure 11, tha a) and b) graphs are not so clera please provide a beller quality of these graphs
Author Response
Dear Editor, Miss Raluca Stefan
Thank you very much for your message of April 17th, 2023 regarding our manuscript. We also would like to thank you for sending the reviewer’s comments to us. Revised manuscript and the list of corrections have been set as per your suggestion. Our incorporation of the reviewer’s suggestion is as follows:
Accepted with minor revisions
- to Fig. 6 for SEM images (It's better to present the dimensions I think it is 200μm for comparison and put out the notes)
Answer: We have added a scale bar.
- SEM images for G1 and G2?
Answer: We have inserted SEM images of G1 and G2
- in Figure 11, tha a) and b) graphs are not so clera please provide a beller quality of these graphs
Answer: It was difficult to work with the old data, but I have made the necessary revisions

Reviewer 4 Report
This paper investigates the vibration damping effect and researching ways to reduce the vibration that occurs during shooting through 3D modeling and simulation, it was possible to evaluate the characteristics and the effect of reducing limb's vibration by manufacturing archery bows and limbs using carbon fiber and glass fiber reinforced composites. Some research results obtained are meaningful to promote the applications of FRP composites in archery bows. However, the authors are encouraged to consider the following minor comments for necessary improvement.
1. Abstract:
(1) Please condense the abstract and give the key research results for the abstract.
(2) The keywords should be further refined.
2. Introduction:
(1) Fiber reinforced composites have been widely used in sports, energy, transportation and civil engineering, owing to the superior mechanical, fatigue and durability performances. The information is important for the readers to understand the materials and applications. The authors should give relevant background in the first paragraph, the following relevant studies can make necessary supplements in the research background, such as “Mechanics of Advanced Materials and Structures, 2023, 30(4):814-834.”, “Composite Structures, 2019, 229: 111427.”, “Engineering Structures, 2023, 274: 115176.”.
(2) Please highlight the innovation of the present paper, especially for the vibration damping.
3. Materials and Methods:
Please give the reason of CFRP for limb and GFRP for stabilizer of archery.
4. Results and discussions:
(1) How many specimens were used in mechanical testing? What about the dispersion of the data?
(2) How to make the the reliable connection between CFRP limb and GFRP stabilizer realized?
(3) Can the authors give the performance improvement ratio of archery after using CFRP limb and GFRP stabilizer?
5. Conclusion:
The conclusion is suggested to be further condensed according to the important findings, give the key results and highlight the innovation of this paper.
Good.
Author Response
Dear Editor, Miss Raluca Stefan
Thank you very much for your message of April 17th, 2023 regarding our manuscript. We also would like to thank you for sending the reviewer’s comments to us. Revised manuscript and the list of corrections have been set as per your suggestion. Our incorporation of the reviewer’s suggestion is as follows:
This paper investigates the vibration damping effect and researching ways to reduce the vibration that occurs during shooting through 3D modeling and simulation, it was possible to evaluate the characteristics and the effect of reducing limb's vibration by manufacturing archery bows and limbs using carbon fiber and glass fiber reinforced composites. Some research results obtained are meaningful to promote the applications of FRP composites in archery bows. However, the authors are encouraged to consider the following minor comments for necessary improvement.
- Abstract:
(1) Please condense the abstract and give the key research results for the abstract.
Answer: We added the sentence “Through testing, the limb and stabilizer that were produced were determined to not fall behind the abilities of the bows currently used by athletes, and they also exhibit a noticeable reduction in vibration.”
(2) The keywords should be further refined.
Answer: We revised the keywords like this: Glass fiber reinforced polymer, Carbon fiber reinforced polymer, Co-curing, Limb, Stabilizer, Archery bow, Vibration damping effect
- Introduction:
(1) Fiber reinforced composites have been widely used in sports, energy, transportation and civil engineering, owing to the superior mechanical, fatigue and durability performances. The information is important for the readers to understand the materials and applications. The authors should give relevant background in the first paragraph, the following relevant studies can make necessary supplements in the research background, such as “Mechanics of Advanced Materials and Structures, 2023, 30(4):814-834.”, “Composite Structures, 2019, 229: 111427.”, “Engineering Structures, 2023, 274: 115176.”.
Answer: We added to the following citation: as “Mechanics of Advanced Materials and Structures, 2023, 30(4):814-834.”, “Composite Structures, 2019, 229: 111427.”, “Engineering Structures, 2023, 274: 115176.”.
(2) Please highlight the innovation of the present paper, especially for the vibration damping.
Answer: The existing BakeliteⓇ limb stabilizer wings showed average vibrations of 500, 2000, and 1200 m·s-2 in the X, Y, and Z axes, respectively, while the 1.5mm GFRP limb stabilizer exhibited significantly reduced vibrations compared to the BakeliteⓇ limb stabilizer, with average vibrations of 280, 1500, and 800 m·s-2 in the X, Y, and Z axes, respectively. On average, vibration decreased by approximately 45%, 25%, and 33% in the X, Y, and Z axes, respectively.
- Materials and Methods: Please give the reason of CFRP for limb and GFRP for stabilizer of archery.
Answer: We added the sentence “The existing recurve bow limbs for athletes all use carbon 3k fabric on the skin surface and carbon U/D on the inside. They are manufactured to the same specifications as those used in actual international competitions. The reason for using GFRP is that it has excellent adhesion and moldability when co-cured with carbon, and the finished product is easy to process with the added benefit of the ability to easily adjust the thickness, color, and material according to the needs.”
- Results and discussions:
(1) How many specimens were used in mechanical testing? What about the dispersion of the data?
Answer: To manufacture the test specimens, 5 were produced for each group, an experiment was carried out.
(2) How to make the reliable connection between CFRP limb and GFRP stabilizer realized?
Answer: CFRP and GFRP are stacked in prepreg form in a mold and molded together in one step using co-curing in an autoclave.
(3) Can the authors give the performance improvement ratio of archery after using CFRP limb and GFRP stabilizer?
Answer: On average, vibration decreased by approximately 45%, 25%, and 33% in the X, Y, and Z axes, respectively, based on objective numerical data.
However, individual differences in other performance aspects were not reflected in the paper. After field tests with domestic and Chinese unemployed team players, it was reported that they did not feel a decrease in performance compared to the previous bows they used, and some even felt that the new bows were better. Currently, some teams are actually using the new bows.
- Conclusion:
The conclusion is suggested to be further condensed according to the important findings, give the key results and highlight the innovation of this paper.
Answer: The existing BakeliteⓇ limb stabilizer wings showed average vibrations of 500, 2000, and 1200 m·s-2 in the X, Y, and Z axes, respectively, while the 1.5mm GFRP limb stabilizer exhibited significantly reduced vibrations compared to the BakeliteⓇ limb stabilizer, with average vibrations of 280, 1500, and 800 m·s-2 in the X, Y, and Z axes, respectively. On average, vibration decreased by approximately 45%, 25%, and 33% in the X, Y, and Z axes, respectively.

Round 2
Reviewer 2 Report
The author has answered all my previous questions and the reviewer suggests checking the article again before submitting.
Reviewer 4 Report
Accept.